# Circulating HPV cDNA in the blood as a reliable biomarker for cervical cancer: A meta-analysis

Yulan Gu[1,2©], Chuandan Wan[3©]*, Jiaming Qiu[4], Yanhong Cui[3], Tingwang Jiang[3], Zhixiang Zhuang[1]*

1 Department of Oncology, the Second Affiliated Hospital of Suzhou University, Suzhou, China, 2 Department of Oncology, the Second People's Hospital of Changshu, Changshu, China, 3 Laboratory of Molecular Biology, Changshu Medical Examination Institute, Changshu, China, 4 Department of Pathology, the Second People's Hospital of Changshu, Changshu, China

© These authors contributed equally to this work.
* cd_wan@163.com (CW); 382561683@qq.com (ZZ)

**Data Availability Statement:** All relevant data are within the paper and its Supporting Information files.

**Funding:** The study was supported by the fund of Key Technologies of Prevention and Control for

## Abstract

The applications of liquid biopsy have attracted much attention in biomedical research in recent years. Circulating cell-free DNA (cfDNA) in the serum may serve as a unique tumor marker in various types of cancer. Circulating tumor DNA (ctDNA) is a type of serum cfDNA found in patients with cancer and contains abundant information regarding tumor characteristics, highlighting its potential diagnostic value in the clinical setting. However, the diagnostic value of cfDNA as a biomarker, especially circulating HPV DNA (HPV cDNA) in cervical cancer remains unclear. Here, we performed a meta-analysis to evaluate the applications of HPV cDNA as a biomarker in cervical cancer. A systematic literature search was performed using PubMed, Embase, and WANFANG MED ONLINE databases up to March 18, 2019. All literature was analyzed using Meta Disc 1.4 and STATA 14.0 software. Diagnostic measures of accuracy of HPV cDNA in cervical cancer were pooled and investigated. Fifteen studies comprising 684 patients with cervical cancer met our inclusion criteria and were subjected to analysis. The pooled sensitivity and specificity were 0.27 (95% confidence interval [CI], 0.24–0.30) and 0.94(95% CI, 0.92–0.96), respectively. The pooled positive likelihood ratio and negative likelihood ratio were 6.85 (95% CI, 3.09–15.21) and 0.60 (95% CI, 0.46–0.78), respectively. The diagnostic odds ratio was 15.25 (95% CI, 5.42–42.94), and the area under the summary receiver operating characteristic curve was 0.94 (95% CI, 0.89–0.99). There was no significant publication bias observed. In the included studies, HPV cDNA showed clear diagnostic value for diagnosing and monitoring cervical cancer. Our meta-analysis suggested that detection of HPV cDNA in patients with cervical cancer could be used as a noninvasive early dynamic biomarker of tumors, with high specificity and moderate sensitivity. Further large-scale prospective studies are required to validate the factors that may influence the accuracy of cervical cancer diagnosis and monitoring.

Major and Infectious Diseases (No. GWZX201604) and Youth Medical Talent Project of Jiangsu (QNRC2016214) to TJ. The funders had no role in designing the study, data collection and analysis, decision to publish, or preparation of the manuscript.

**Competing interests:** The authors have declared that no competing interests exist.

## Introduction

Human papillomavirus (HPV) is a type of papillomavirus that infects human skin and mucosa squamous epithelial cells. HPVs are DNA double-stranded spherical small viruses with a diameter of about 55 nm. The HPV genome contains approximately 7900 bases and can be divided into three functional regions [1]. The proteins E6 and E7, encoded by the early genes of HPV, can inhibit the functions of p53 and pRh in normal cervical epithelial cells and cause abnormal proliferation of cancerous cells, resulting in the development of genital warts and atypical proliferation of epithelial cells [2]. The immune system of most patients can eliminate HPV within approximately 9–16 months after infection. However, persistent infection by some high-risk HPVs, particularly HPV16 and HPV18, may lead to cervical cancer [3–5].

Cervical cancer is the fourth most common cancer among women worldwide. However, 85% of cases occur in developing countries [6]. Cervical cancer is now relatively uncommon in high-income countries owing to the introduction of HPV screening programs and HPV vaccines, which have led to a 70% decrease in the incidence and mortality rates of cervical cancer over past several decades [7]. Despite major advances in detection and prevention, an estimated 530,000 cases were recorded, and nearly 90% of 270,000 deaths occurred in middle- and low-income developing countries in 2012 [8]. There is still a need for minimally invasive and specific tests for HPV-induced cancer.

Recent progress in the analysis of blood samples for circulating tumor cells or cell-free circulating tumor DNA (ctDNA) has shown that liquid biopsies may have potential applications in the detection and monitoring of cancer [9–11]. Similarly, in a study on cervical cancer, HPV cDNA has become a major focus, providing a strong basis for early diagnosis and prognosis in cervical cancer [10, 12, 13]. Cervical cancer is typically caused by high-risk HPVs (hrHPVs), primarily genotypes 16 and 18 [4]. hrHPVs linearize DNA for integration into the cervical host genome and induce the expression of E6 and E7 genes, which are involved in the oncogenesis of cervical cancer [14, 15]. Cervical cancer cells and HPV cDNA harbor genomic rearrangements that can be released into the patient's peripheral blood. From a diagnostic monitoring viewpoint, the consistent presence of HPV cDNA in the blood of patients with cervical cancer can be used as a tumor marker. Although the mechanism mediating this phenomenon is unclear, the presence of such HPV cDNA in cervical cancer shows some diagnostic value. Interestingly, some studies have shown that circulating HPV cDNA acts as a tumor DNA marker in patients with primary tumors caused by HPV infection [10]. Many recent studies have focused on ctDNA in cervical cancer; however, the exact relationships are still unclear [12, 16–18].

Accordingly, in this study, we performed a comprehensive analysis of the precise value of HPV cDNA for the diagnosis of cervical cancer.

## Materials and methods

### Protocol

The complete protocol is available here: http://dx.doi.org/10.17504/protocols.io.8t4hwqw.

### Search strategy

This meta-analysis was conducted following the criteria of Preferred Reporting Items for Systematic Review and Meta Analyses [19]. A literature search was systematically performed using PubMed, Embase, Cochrane Library, and WANFANG medicine online databases for all relevant articles without language or regional limitations. No limitations were set with regard to the start date for publication, and the search ended on March 18, 2019. The following search

terms were used: "cervical cancer AND HPV cDNA", "cervical cancer AND ctDNA", "cervix cancer AND ctDNA", "cervical carcinoma AND ctDNA" OR "circulating DNA AND cervical cancer". Various alterations in spelling and abbreviations were also used as search terms. Titles and abstracts were carefully screened for relevance, and duplicates were removed. The full text of each report that met the preliminary criteria was retrieved and assessed for inclusion into this meta-analysis.

## Inclusion and exclusion criteria

In this meta-analysis, eligible studies were selected according to these following inclusion criteria: (1) evaluated the diagnostic accuracy of quantitative analysis of HPV cDNA in cervical cancer; (2) the diagnostic value of HPV cDNA in cervical cancer was reported or could be calculated from the published data; (3) full text and all data could be retrieved and were available; (4) the techniques and target genes were clearly stated in the articles; (5) studies included at least 10 patients with cervical cancer and relevant negative controls. When the same patient population was used in several studies, only the most recent was included.

The exclusion criteria were as follows: (1) the diagnostic or prognostic value could not be deduced from incomplete data in the studies provided; (2) repeated studies from the same study group; (3) sample size less than 10; (4) data only from experiments based on cell lines; (5) studies published in languages other than English.

## Quality assessment

Two reviewers (CD Wan and YL Gu) independently reviewed and evaluated all eligible studies according to the Newcastle-Ottawa scale [20]. In case of disagreement, the decision was made by a third researcher, and disagreement was settled through discussion. The data extracted from the basic feature table included authors' names, country, sample type, detection method, numbers of experimental and control groups, and analysis indicators. The outcome indicators included positives, false positives, false negatives, true negatives, sensitivity, and specificity. To assess the methodological quality of each study and potential risk of bias, QUADAS-2 Guidelines were used to evaluate the quality of all articles that met the inclusion criteria [21].

## Statistical analysis

We used standard methods recommended for meta-analysis of diagnostic test evaluations [19]. The meta-analysis was carried out with Meta-DiSc 1.4 and STATA 14.0 statistical software. The sensitivity was defined as the proportion of patients with HPV cDNA presence among all patients confirmed as having cervical cancer. The specificity was defined as the proportion of patients with negative HPV cDNA detection among all negative control volunteers without cervical cancer. The positive likelihood ratio (PLR) was calculated as sensitivity/(1 – specificity), whereas the negative likelihood ratio (NLR) was calculated as 1 –sensitivity/specificity. DOR was calculated as PLR / NLR and was used as an indication of how much greater the chance was of having cervical cancer for patients with HPV cDNA presence than for those without HPV cDNA. These indicators were summarized using a bivariate meta-analysis model, and the threshold effect was determined by receiver operative characteristic (ROC) curve and Spearman correlation analyses; $P$ values of less than 0.05 indicated a significant threshold effect. Heterogeneity between studies was analyzed by chi-squared and $I^2$ tests; a $P$ value of less than 0.1 or an $I^2$ higher than 50% indicated the existence of significant heterogeneity [22]. Meta-regression analysis was performed to explore the sources of heterogeneity. Deek's funnel plot asymmetry test was used to test whether there was publication bias [23]. All

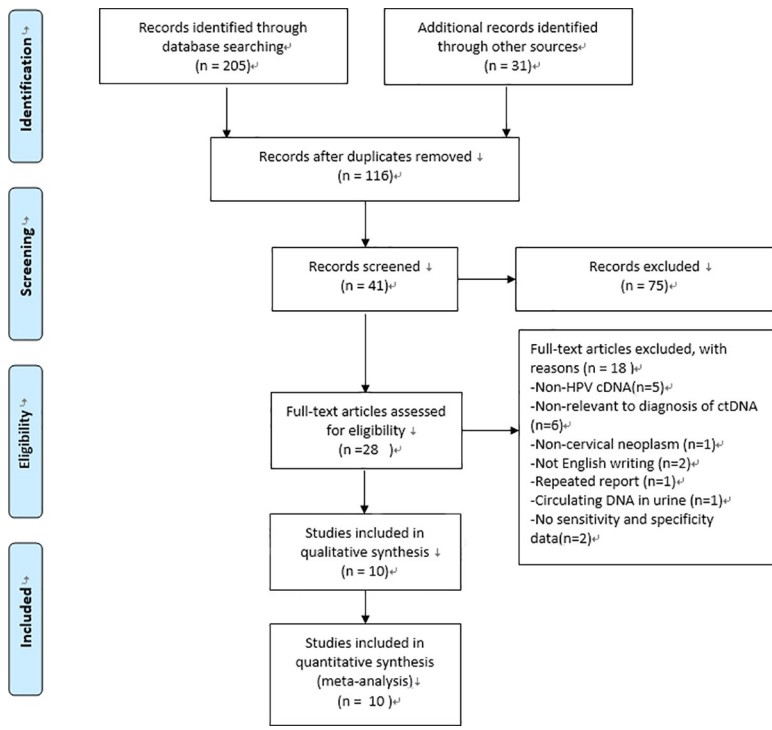

**Fig 1. Flow chart of the enrolled studies.**

statistical tests were two-sided, and results with *P* values of less than 0.05 were considered statistically significant.

## Results

### Study selection process

The initial search retrieved a total of 236 studies. As shown in Fig 1, 10 studies were eligible for review after carefully screening and rechecking. All relevant characteristics of these studies are summarized in Table 1. In total, 684 patients with cervical cancer were evaluated in these

**Table 1. Main characteristics of all the studies enrolled the meta-analysis.**

| No. | Study | year | region | method | TP | FP | FN | TN | Sample source | Sample time | sensitivity | specificity | scores |
|-----|-------|------|--------|--------|----|----|----|----|---------------|-------------|-------------|-------------|--------|
| 1 | Pornthanakasem W[10] | 2001 | Thailand | qPCR | 6 | 0 | 44 | 20 | plasma | BT | 36.00% | 100.00% | 7 |
| 2 | Dong SM[24] | 2002 | America | qPCR | 13 | 1 | 219 | 59 | plasma | C | 48.70% | 98.33% | 7 |
| 3 | Hsu KF[25] | 2003 | Taiwan | qPCR | 27 | 0 | 85 | 40 | serum | BT | 45.2% | 88.60% | 6 |
| 4 | Sathish N[26] | 2004 | India | PCR+RFLP | 8 | 0 | 50 | 40 | plasma | BT | 48.2% | 100.00% | 8 |
| 5 | Yang HJ[27] | 2004 | HongKong | qPCR | 34 | 17 | 34 | 94 | plasma | BT | 50% | 84.68% | 8 |
| 6 | Wei YC[28] | 2007 | Taiwan | Nested qPCR | 11 | 0 | 6 | 6 | plasma | BT | 64.70% | 100.00% | 5 |
| 7 | Jaberipour M[29] | 2011 | Iran | qPCR | 19 | 8 | 62 | 80 | plasma | BT | 23.5% | 90.91% | 8 |
| 8 | Campitelli M[30] | 2012 | France | DIPS-PCR | 13 | 0 | 3 | 20 | serum | BT | 81.25% | 100.00% | 7 |
| 9 | Jeannot E[31] | 2016 | France | ddPCR | 39 | 0 | 8 | 18 | serum | BT | 83.00% | 100.00% | 6 |
| 10 | Kang Z[17] | 2017 | America | ddPCR | 19 | 0 | 2 | 45 | serum | C | 90.48% | 100.00% | 7 |

Sample time:BT, before treatment; C, combined

studies published between 2001 and 2018. Among these studies, 6 enrolled patients from Asian countries/areas (one from Hong Kong, one from Thailand, one from India, one from Iran, and one from Taiwan). Additionally, two studies were performed in France, and two were performed in America. Numerous review papers and duplicates between the literature databases were excluded.

## Review of eligible studies

The 10 eligible studies with data regarding the diagnostic value of HPV cDNA in cervical cancer are shown in Table 1. From these studies, 263 patients with cervical cancer were evaluated before treatment, and 421 patients were evaluated when undergoing treatment or after treatment. Patients with primary or metastatic cervical cancer with a TNM stage of I–IV who received surgery, chemotherapy, radiotherapy, or targeted therapy were included. The types of cervical cancer were squamous and adenomatous (approximate ratio of 4:1), as shown in S1 Table.

## Quality assessment

The quality score of all studies was 6 to 8 points, with an average of 6.9 (Table 1). A quality assessment of the eligible studies was performed using QUADAS-2 (Fig 2). The included 10 studies were assessed using RevMan 5.3 software, and most of studies showed moderately low or unclear risk of bias. Two studies [10, 30] increased the risk of bias owing to a lack of patient selection. Two studies [24,31] did not mention the use of a blinding method or reference standard, which may have resulted in an unknown risk of bias in the meta-analysis.

## Detection of HPV cDNA and probes

Polymerase chain reaction (PCR) was mainly applied to detect HPV cDNA in the studies included in this analysis. Two studies [10, 31] used Taqman PCR. Additionally, two studies

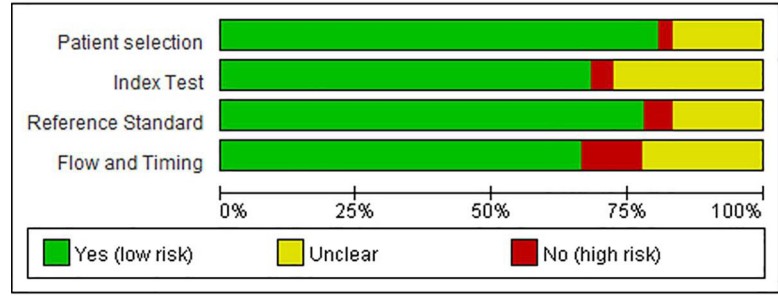

### Risk of bias

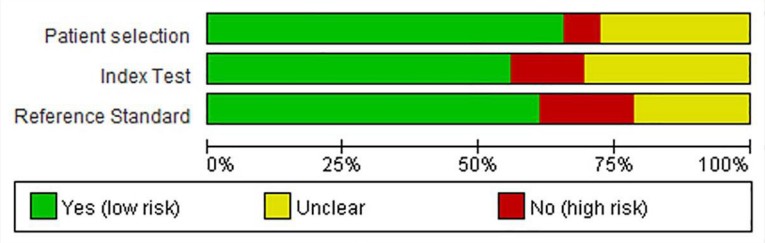

### Applicability concerns

**Fig 2. Quality assessment of the included studies according to QUADAS-2.**

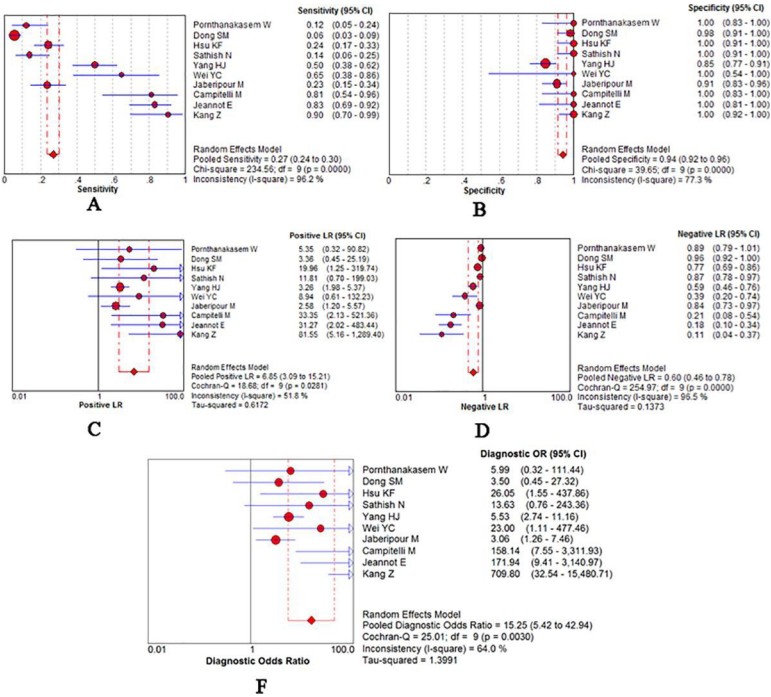

**Fig 3. Diagnostic accuracy forest plots.** (A) Forest plots of pooled sensitivity. (B) Forest plots of pooled specificity. (C) Forest plots of PLR. (D) Forest plots of NLR. (F)Forest plots of pooled DOR.

[17, 28] used droplet digital PCR (ddPCR), and one study [30] used methylation-specific (MSP) PCR and one study [28] used nested PCR. Six of the studies extracted ctDNA from plasma, and the other four studies extracted DNA from serum (Table 1). For the HPV cDNA the probes used the different studies were not exactly the same, as showed in S2 Table.

## HPV cDNA diagnostic accuracy in cervical cancer

All of 10 studies were pooled into meta-analysis of diagnostic accuracy. The Spearman correlation coefficient was 0.276 ($P>0.05$), suggesting that there was no threshold effect. Accordingly, heterogeneity owing to non-threshold effects was assessed with $Q$ tests and $I^2$ statistics. There was significant heterogeneity in the pooled sensitivity($I^2 = 96.2\%$, $P<0.001$) and specificity ($I^2 = 77.3\%$, $P<0.001$); Thus a random effects model would be applied to analyze the diagnostic parameters. As presented in Fig 3, the overall pooled sensitivity and specificity were 0.27(95% CI 0.24–0.30) and 0.94(95%CI 0.92–0.96), respectively. The overall pooled positive likelihood ratio (PLR) and negative likelihood ratio (NLR) were 6.85 (95%CI 3.09–15.21) and 0.60 (95% CI 0.46–0.78), respectively. The pooled diagnostic odd ratio (DOR) was 15.25 (95%CI 5.42–42.94). The summary receiver operator characteristic curve (SROC) was presented in Fig 4; the area under the SROC curve AUC was 0.94 (95%CI 0.89–0.99).

## Subgroup analysis and meta-regression

Subgroup analysis was performed to explain the source of the significant heterogeneity in the diagnostic analysis. These different parameters in all of included studies were conducted including sample source (serum versus plasma), sample time (before treatment versus others), race or region (Asian versus Caucasian), patient number(≥50 cases versus <50 cases), and detection method (qPCR vs MSP and ddPCR). These diagnostic parameters of subgroups

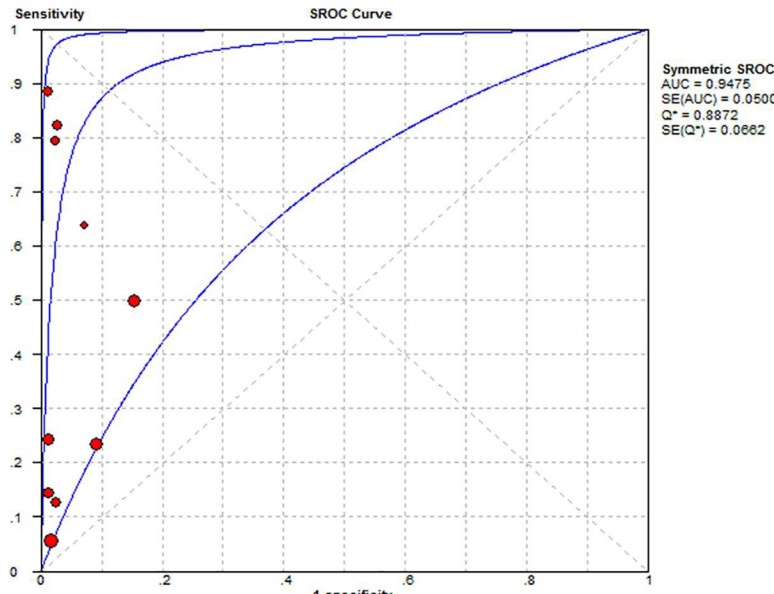

**Fig 4. Summary receiver operating characteristic plot for the pooled studies diagnosis.**

were showed in Table 2. Meta-regression based on those five factors were applied to investigate the source of heterogeneity. As showed in Table 3, the source and race or region factors showed significantly influence on heterogeneity of universal diagnostic value (*P*<0.05).

## Sensitivity analysis

To further explore the heterogeneity of the included studies, a sensitivity analysis was conducted by removing individual studies. As shown in Fig 5, no outlier study was identified, and the results were considerable stable and reliable.

**Table 2. Results of subgroups analysis.**

| Subgroup | Sensitivity(95% CI) | Specificity(95% CI) | PLR(95% CI) | NLR(95% CI) | DOR(95% CI) |
|---|---|---|---|---|---|
| **Source** | | | | | |
| Plasma | 0.18(0.15–0.22) | 0.92(0.89–0.95) | 3.25(2.19–4.83) | 0.80(0.66–0.96) | 4.76(2.86–7.91) |
| Serum | 0.50(0.43–0.57) | 1.00(0.97–1.00) | 36.12(9.10–143.24) | 0.25(0.04–1.61) | 139.15(31.72–610.40) |
| **Method** | | | | | |
| qPCR | 0.21(0.18–0.25) | 0.94(0.91–0.96) | 4.70(2.35–9.40) | 0.74(0.60–0.91) | 8.70(3.41–22.22) |
| MSP and ddPCR | 0.83(0.71–0.91) | 1.00(0.97–1.00) | 32.29(4.64–224.73) | 0.19(0.11–0.32) | 165.21(20.21–1350.4) |
| **Race or Region** | | | | | |
| Mongolian | 0.27(0.23–0.32) | 0.92(0.88–0.95) | 3.37(2.26–5.03) | 0.78(0.68–0.89) | 5.14(3.07–8.60) |
| Caucasian | 0.27(0.22–0.32) | 0.99(0.96–1.00) | 18.65(4.04–86.14) | 0.26(0.01–7.97) | 76.54(5.92–989.35) |
| **Time** | | | | | |
| Before treatment | 0.35(0.31–0.40) | 0.93(0.89–0.95) | 5.40(2.56–11.38) | 0.63(0.50–0.79) | 11.54(4.30–30.98) |
| Under- or after treatment | 0.13(0.10–0.17) | 0.99(0.95–1.00) | 14.51(0.55–382.34) | 0.34(0.00–175.98) | 43.82(0.24–799.36) |
| **Patient Number** | | | | | |
| <50 case | 0.56(0.49–0.62) | 0.92(0.88–0.95) | 12.78(2.74–59.60) | 0.33(0.15–0.75) | 40.37(6.17–265.04) |
| ≥50 case | 0.14(0.11–0.17) | 0.96(0.93–0.98) | 3.78(1.55–9.23) | 0.86(0.75–0.98) | 4.30(1.83–10.10) |
| Overall | 0.27(0.24–0.30) | 0.94(0.92–0.96) | 6.85(3.09–15.21) | 0.60(0.46–0.78) | 15.25(5.42–42.94) |

**Table 3. Meta regression of diagnostic value.**

| parameter | Coef | SE | RDOR(95%CI) | P |
|---|---|---|---|---|
| Source | -3.414 | 0.7992 | 0.03(0.00–0.22) | 0.0037 |
| Method | -2.620 | 1.4951 | 0.07(0.00–2.50) | 0.1232 |
| Race or region | -2.536 | 0.7686 | 0.08(0.01–0.49) | 0.0131 |
| Time | -1.483 | 1.6317 | 0.23(0.00–10.75) | 0.3935 |
| Patient number | 1.569 | 1.6562 | 4.80(0.10–241.01) | 0.3751 |

## Publication bias

We applied Deeks' funnel plot asymmetry tests to estimate the publication bias of the included studies. As shown in Fig 6, the regression line was nearly vertical, confirming the lack of significant publication bias across the overall enrolled studies (P = 0.49).

## Discussion

Although pathological examination is the gold standard of clinical tumor treatment, obtaining such specimen without interruption directly from tumors is a difficult procedure and cannot reflect tumor dynamic changes after treatment [32]. Cancers are known to shed tumor cell DNA into the blood stream [33], and examining the levels and mutations in ctDNA can provide almost real-time information regarding tumor status, which is called "liquid biopsy". Liquid biopsy has the potential to improve post-treatment surveillance by following subtle changes in tumor cfDNA and has recently been extensively investigated as a potential new diagnostic technique [32, 34].

Despite major advances in early detection, including Pap smears and co-human papillomavirus testing, cervical cancer is the fourth leading cause of cancer-related death in women worldwide [35]. There is an urgent need for a minimally invasive and specific test for disease monitoring. Persistent infection and integration of HPV into the cell genome are the first causes of most cervical cancers. After integration, its HPV gene behaves the same as other functional genes of human chromosome [35]. The proliferation or apoptosis of cervical cancer

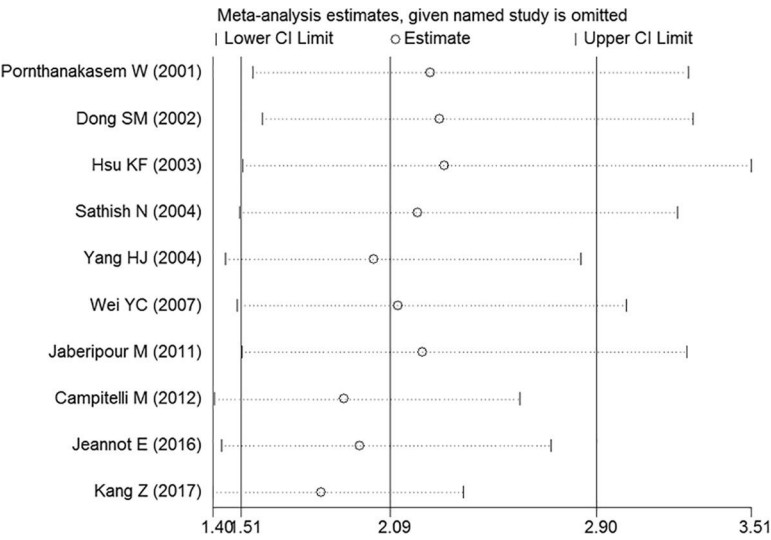

**Fig 5. Sensitivity analysis of the overall pooled study.**

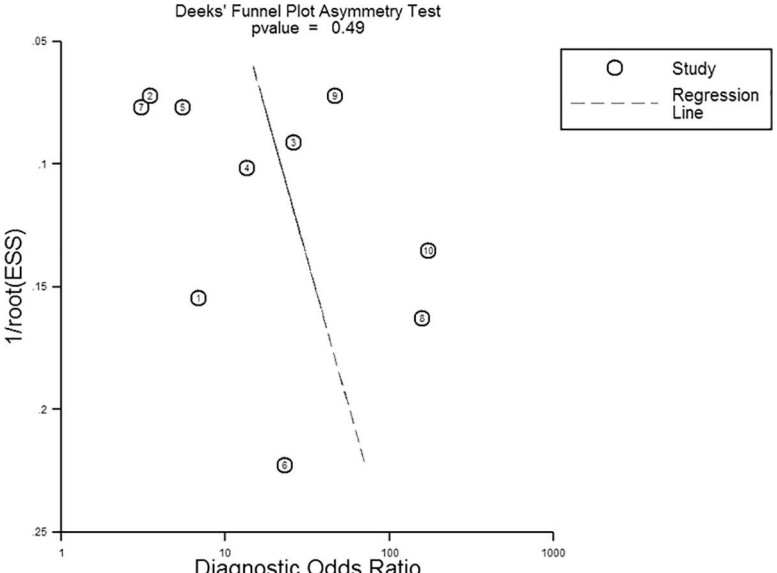

**Fig 6. Deek's funnel plot to assess publication bias.** ESS, effective sample size.

cells will release ctDNA into the peripheral blood circulation system, including HPV DNA. As one of ctDNAs in cervical cancer, circulating HPV DNA (HPV cDNA) can be widely evaluated using liquid biopsies for detecting cancer and monitoring disease [36].

Many previous meta-analyses have reported that the diagnostic accuracy of quantitative analysis of ctDNA is superior to conventional biomarkers for the diagnosis of several cancers, including ovarian cancer [22], gastric cancer [32], lung cancer [37], and colon cancer [38]. To the best of our knowledge, this is the first meta-analysis exploring HPV cDNA in patients with cervical cancer. Meta-analysis can overcome the problem of small sample size and inadequate statistical power in genetic studies of complex traits and provide more reliable results than single case-control studies [39]. Because the relationship between HPV cDNA and cervical cancer is still unclear, we performed a comprehensive analysis of the clinical utility of HPV cDNA in the diagnosis of patients with cervical cancer.

This meta-analysis combined the outcomes of 684 patients with cervical cancer from 10 individual studies, investigating the diagnostic values of HPV cDNA. From the 10 studies, the pooled sensitivity and specificity were 0.27 (95% CI, 0.24–0.30) and 0.94 (95% CI, 0.92–0.96), respectively. LRs of greater than 10 or less than 0.1 indicate large and often conclusive shifts from pretest to post-test probability [34]. In this meta-analysis, the overall pooled PLR and NLR were 6.85 (95% CI, 3.09–15.21) and 0.60 (95% CI, 0.46–0.78), respectively. This result indicated that patients with cervical cancer had approximately 7 times greater chance of being HPV cDNA positive than normal controls, with an error rate of approximately 60% when the true negative was determined in the HPV cDNA negative test. The pooled DOR was 15.25 (95% CI, 5.42–42.94), which indicated a relatively high accuracy of HPV cDNA in cervical cancer. Summary ROC (SROC) can be applied to summarize overall test performance, and the area under the SROC curve (AUC) was 0.94 (95% CI, 0.89–0.99), suggesting that HPV cDNA in the plasma or serum of patients with cervical cancer had excellent accuracy for diagnosing cervical cancer. Because significant heterogeneity existed, if relatively accurate diagnostic parameters were achieved, subgroup analysis would be needed to analyze the source. Subgroup analyses revealed that the heterogeneity of sensitivity could be related to the source of the specimen (e.g., plasma versus serum), the region (e.g., Mongolian versus Caucasian), the time of

specimen collection (e.g., before treatment versus after treatment), the number of patients (e.g., less than 50 versus greater than or equal to 50), and the method of analysis (e.g., quantitative PCR versus MSP-PCR and ddPCR). Most studies employed a qPCR method that demonstrated relatively high specificity but low sensitivity. With qPCR it is difficult to detect very small amounts of circulating nucleic acids in blood. Over time, more accurate diagnostic parameters were obtained by ddPCR. We found that MSP-PCR and ddPCR were more accurate for detecting HPV cDNA in patients than qPCR. With the application of new detection methods, higher sensitivity and specificity had been obtained. In particular, the application of ddPCR in liquid biopsy greatly improves the diagnostic value of HPV cDNA [36]. However, statistical regression data showed that all these differences between subgroups were not statistically significant ($P > 0.05$). Taken together, these results indicated that the study design did not substantially affect the diagnostic accuracy. Heterogeneity may have been caused by other factors, such as patient age, tumor type, tumor size, TNM stage, and differences in the experimental protocols, which could not be analyzed in the current study because of loss of data or unrecognizable details. Therefore, further studies with large sample sizes and more details, e.g., race, specimen features, and tumor properties, are needed to confirm these findings.

Cervical cancer differs from other cancers because HPV infection is a crucial step in tumorigenesis, accounting for 99.7% of cervical cancer cases. HPV16 and HPV18 are the two most important serotypes, identified in more than 70% of cervical carcinomas worldwide [40]. Specific changes in circulating nucleics with regard to oncogenes, tumor-suppressor mutations, microsatellite alterations, and hypermethylation can be similarly detected. Although the 10 studies included in this meta-analysis had very high specificity, there was uneven sensitivity. The pooled results indicated that there was significant heterogeneity in sensitivity that could impact diagnostic accuracy. The Spearman correlation coefficient was 0.276 ($P > 0.05$), suggesting that the threshold effect was not the source of heterogeneity. Because the size of HPV cDNA fragments is generally approximately 200 bp [41], PCR primer pairs that target shorter DNA fragments may identify more patients with detectable HPV DNA. With more primer pairs, further increases in detection rates may be possible. Other influencing factors, such as patient number, specimen extraction time, region, and specimen source, may also influence these parameters; however, these differences were not statistically significant. In addition, publication bias was also not significant, indicating that the results of this meta-analysis were reliable and credible.

There were several limitations to this meta-analysis. First, the sensitivities of the included studies varied widely. Different gene detection methods could have led to major differences. Therefore, significant heterogeneity between studies could not be avoided. The unique characteristics of HPV cDNA limit its sensitivity as a diagnostic indicator, and more sensitive and accurate detection techniques may need to be applied. Although subgroup and regression analyses were performed to explore the sources of heterogeneity, the results of these analyses explained few effectors. Second, some studies with limited patient numbers and controls were included in this meta-analysis, reducing the effectiveness of the combined statistical analysis. Relatively few papers on HPV cDNA in patients with cervical cancer have been published. Third, owing to the nature of our research, selected bias and incomplete searches could have occurred. Finally, different probes used for PCR of HPV cDNA may have result in a potential source of bias.

## Conclusions

Despite some limitations, this meta-analysis clearly indicated that HPV cDNA detection may be a very specific, but relatively sensitive test in patients with cervical cancer. Our findings

provided reliable evidence that HPV cDNA was a promising potential biomarker for the diagnosis of cervical cancer. Of course, to obtain a more accurate statistical data analysis, additional studies with larger sample sizes from patients of different ethnicities will be necessary in the future.

## Supporting information

**S1 Table. All information extracted from eligible studies.**
(XLSX)

**S2 Table. The primers of the detected HPV cDNA in 10 enrolled studies.**
(DOCX)

**S3 Table. PRISMA checklist.**
(DOC)

**S1 Fig. The full search strategy and search terms used for PubMed database.**
(DOCX)

## Author Contributions

**Conceptualization:** Chuandan Wan, Jiaming Qiu.

**Data curation:** Yulan Gu, Chuandan Wan, Jiaming Qiu.

**Formal analysis:** Yulan Gu, Chuandan Wan, Jiaming Qiu.

**Investigation:** Tingwang Jiang.

**Methodology:** Tingwang Jiang.

**Supervision:** Zhixiang Zhuang.

**Validation:** Yanhong Cui.

**Visualization:** Yanhong Cui.

**Writing – original draft:** Chuandan Wan.

**Writing – review & editing:** Yulan Gu, Chuandan Wan.

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
