## [Editor Report · Decision Letter 0]

24 Oct 2019

PONE-D-19-27050

Diagnostic value of circulating tumor DNA as an effective biomarker in cervical cancer: a meta-analysis

PLOS ONE

Dear Mr. Wan,

Thank you for submitting your manuscript to PLOS ONE. After careful consideration, we feel that it has merit but does not fully meet PLOS ONE’s publication criteria as it currently stands. Therefore, we invite you to submit a revised version of the manuscript that addresses the points raised during the review process.

We would appreciate receiving your revised manuscript by Dec 08 2019 11:59PM. To enhance the reproducibility of your results, we recommend that if applicable you deposit your laboratory protocols in protocols.io, where a protocol can be assigned its own identifier (DOI) such that it can be cited independently in the future. For instructions see: http://journals.plos.org/plosone/s/submission-guidelines#loc-laboratory-protocols

We look forward to receiving your revised manuscript.

Kind regards,

Peter van Dam

Academic Editor

PLOS ONE

Journal Requirements:

3. We noticed minor instances of text overlap with the following previous publication(s), which need to be addressed:

(1) https://linkinghub.elsevier.com/retrieve/pii/S0090825817308946

(2) http://www.medsci.org/v10p0981.htm

(3) https://www.sciencedirect.com/science/article/abs/pii/S1386653219300502?via%3Dihub

The text that needs to be addressed involves the Discussion section, specifically (1) the first paragraph, (2) the third paragraph and (3) the second to last page of the Discussion section.

In your revision please ensure you cite all your sources (including your own works), and quote or rephrase any duplicated text outside the methods section. Further consideration is dependent on these concerns being addressed.

4. At this time, we ask that you please provide the full search strategy and search terms used for at least one database as Supplementary information.

NO

a) Please provide an amended Funding Statement that declares *all* the funding or sources of support received during this specific study (whether external or internal to your organization) as detailed online in our guide for authors at http://journals.plos.org/plosone/s/submit-now.  

b) Please state what role the funders took in the study.  If any authors received a salary from any of your funders, please state which authors and which funder. If the funders had no role, please state: "The funders had no role in study design, data collection and analysis, decision to publish, or preparation of the manuscript."

Additional Editor Comments:

Before sending this paper to the reviewers I feel the authors should rewrite the present paper. It contains valuable information but I feel it is non-sense to perform a metanalysis combining HPV and non-HPV tests. The metaanalysis should mainly focus on HPV cDNA. The studies looking at non-HPV DNA (Table 1: studies 5,11,14,15) should be analyzed separately. For the HPV cDNA the probes used in the different studies shoudl be mentioned in a Table
---

## [Author Response · Author response to Decision Letter 0]

18 Nov 2019

We appreciate your suggestion on focusing on the data from HPV cDNA in our meta-analysis. Based on the findings from our original manuscript, we believe that, after integration into the human chromosome, HPV cDNA behaves like other functional genes. Subgroup analysis also showed that there were no significant differences between HPV cDNA and ctDNA of other genes. However, too many miscellaneous ctDNAs may potentially result in heterogeneity that is incompatible with the setup for meta-analysis. Therefore, we have excluded studies on non-HPV cDNA and solely focused on studies including HPV cDNA. The tables and figures have also been replaced in the revised manuscript.

---

## [Decision Letter · Decision Letter 1]

27 Dec 2019

PONE-D-19-27050R1

Circulating HPV cDNA in the blood as a reliable biomarker for cervical cancer: a meta-analysis

PLOS ONE

Dear Mr. Wan,

Thank you for submitting your manuscript to PLOS ONE. After careful consideration, we feel that it has merit but does not fully meet PLOS ONE’s publication criteria as it currently stands. Therefore, we invite you to submit a revised version of the manuscript that addresses the points raised during the review process.

We would appreciate receiving your revised manuscript by Feb 10 2020 11:59PM. To enhance the reproducibility of your results, we recommend that if applicable you deposit your laboratory protocols in protocols.io, where a protocol can be assigned its own identifier (DOI) such that it can be cited independently in the future. For instructions see: http://journals.plos.org/plosone/s/submission-guidelines#loc-laboratory-protocols

We look forward to receiving your revised manuscript.

Kind regards,

Peter van Dam

Academic Editor

PLOS ONE

Additional Editor Comments (if provided):

Cfr supra

Reviewers' comments:

Reviewer's Responses to Questions

**Comments to the Author**

1. If the authors have adequately addressed your comments raised in a previous round of review and you feel that this manuscript is now acceptable for publication, you may indicate that here to bypass the “Comments to the Author” section, enter your conflict of interest statement in the “Confidential to Editor” section, and submit your "Accept" recommendation.

Reviewer #1: (No Response)

2. Is the manuscript technically sound, and do the data support the conclusions?

Reviewer #1: Partly

3. Has the statistical analysis been performed appropriately and rigorously? 

Reviewer #1: No

4. Have the authors made all data underlying the findings in their manuscript fully available?

Reviewer #1: No

5. Is the manuscript presented in an intelligible fashion and written in standard English?

Reviewer #1: Yes

6. Review Comments to the Author

Reviewer #1: This manuscript reports meta-analysis results for HPV cDNA in the blood as a diagnostic biomarker for cervical cancer. I have below comments.

For Table 1, please add the number of cases, non-cases, TP, FP, FN, TN and cut-off values (if any) for each study. The sensitivities of No. 1, 2 and 4 studies are different from those reported in Table S1.

For quality assessment, please provide a table to report the QUADAS assessment scores for each included study individually.

Page 10, under Quality assessment, the 2nd line “The included 15 studies…” should be “The included 10 studies...”

In Figure 3, it is not clear if random effects model was used for pooled sensitivity or specificity.

In Figure 3, please add meta-analysis analysis for DOR.

In subgroup analysis, race is a factor, but in Table 2A, no race is included. The term of race or region and their contents should be consistent through the text and tables.

What are the estimates in Figure 5? For sensitivity analysis please include pooled results for each of major outcomes (sensitivity, specificity, and DOR).

Page 18, in Discussion, the statement, “Most studies employed a qPCR method that demonstrated relatively high sensitivity and specificity”, does not reflect the truth. From Tables 1 and S1 (even though the numbers are not consistent between these two tables), those studies with qPCR method show very low sensitivities.

From Table 1, it seems DIPS-PCR and ddPCR would give high sensitivity but not other method. But in Discussion, authors don’t think the test methods would contribute heterogeneity, which is not convincing.

7. PLOS authors have the option to publish the peer review history of their article (what does this mean?). If published, this will include your full peer review and any attached files.

Reviewer #1: No

---

## [Author Response · Author response to Decision Letter 1]

9 Jan 2020

Thank you for your time and effort in reviewing our manuscript and providing us with comprehensive and valuable feedback.

Please find below the clarifications for the concerns raised:

Review Comment: For Table 1, please add the number of cases, non-cases, TP, FP, FN, TN and cut-off values (if any) for each study. The sensitivities of No. 1, 2 and 4 studies are different from those reported in Table S1.

Clarifications: The number of TP, FP, FN, TN for each study has been added in Table1. Meanwhile some unimportant data have been deleted from Table 1. The sensitivities of No.1, 2 and 4 studies in Table S1 also have been corrected. 

Review Comment: For quality assessment, please provide a table to report the QUADAS assessment scores for each included study individually.

Clarifications: QUASAS assessment scores were added to Table 1.

Review Comment: Page 10, under Quality assessment, the 2nd line “The included 15 studies…” should be “The included 10 studies...”

Clarifications: In the last revision, the original 15 papers were included in this meta-analysis. According to the requirements, the theme was re-locked on HPV cDNA. The original 15 articles were reduced to 10. We forgot to revise them here. Now it has been corrected.

Review Comment: In Figure 3, it is not clear if random effects model was used for pooled sensitivity or specificity. In Figure 3, please add meta-analysis analysis for DOR.

Clarifications: In Figure 3, both sensitivity and specificity analysis are obtained under the random effects mode. DOR results have also been added to Table 3.

Review Comment: In subgroup analysis, race is a factor, but in Table 2A, no race is included. The term of race or region and their contents should be consistent through the text and tables.

Clarifications: The term of race or region and their contents have been checked consistently through the text and tables.

Review Comment: What are the estimates in Figure 5? For sensitivity analysis please include pooled results for each of major outcomes (sensitivity, specificity, and DOR).

Clarifications: Sensitivity, specificity, and DOR aren’t generally provided in sensitivity analysis of meta review literatures (including sensitivity analysis of meta literature in PLoS One and other journals). The sensitivity analysis here is to examine the effect of a single study on the total combined effect (estimates and their 95%CI). Figure 5 showed that the results were relatively stable and reliable.

Review Comment: Page 18, in Discussion, the statement, “Most studies employed a qPCR method that demonstrated relatively high sensitivity and specificity”, does not reflect the truth. From Tables 1 and S1 (even though the numbers are not consistent between these two tables), those studies with qPCR method show very low sensitivities.

Clarifications: The original sentence has been revised to: “Most studies employed a qPCR method that demonstrated relatively high specificity but low sensitivity. The sensitivity of qPCR is difficult to detect very small amounts of circulating nucleic acids in blood.” It is true that MSP-PCR and ddPCR were more accurate for detecting HPV cDNA in patients than qPCR. The reasons for these results are also discussed in text.

Review Comment: From Table 1, it seems DIPS-PCR and ddPCR would give high sensitivity but not other method. But in Discussion, authors don’t think the test methods would contribute heterogeneity, which is not convincing.

Clarifications: From the perspective of raw data, differences in methodology can indeed bring about greater heterogeneity, but meta-analysis shows that the test methods do not contribute statistically significant heterogeneity. In discussion, we suggested some possible causes of heterogeneity. Of cause, further studies with large sample sizes and more details, e.g., race, specimen features, and tumor properties, are needed to confirm these suggestions. 

The revised parts of the manuscript have been highlighted in yellow. Thank you for your consideration. I look forward to your response.

---

## [Editor Report · Decision Letter 2]

13 Jan 2020

Circulating HPV cDNA in the blood as a reliable biomarker for cervical cancer: a meta-analysis

PONE-D-19-27050R2

Dear Dr. Wan,

We are pleased to inform you that your manuscript has been judged scientifically suitable for publication and will be formally accepted for publication once it complies with all outstanding technical requirements.

With kind regards,

Peter van Dam

Academic Editor

PLOS ONE

Additional Editor Comments (optional):

Just change: With qPCR it is difficult to detect .... (discussion)
---

## [Editor Report · Acceptance letter]

23 Jan 2020

PONE-D-19-27050R2 

Circulating *HPV* cDNA in the blood as a reliable biomarker for cervical cancer: a meta-analysis 

Dear Dr. Wan:

I am pleased to inform you that your manuscript has been deemed suitable for publication in PLOS ONE. Congratulations! Your manuscript is now with our production department. 

With kind regards,

on behalf of

Dr. Peter van Dam 

Academic Editor

PLOS ONE